# Role of Splicing Regulatory Elements and In Silico Tools Usage in the Identification of Deep Intronic Splicing Variants in Hereditary Breast/Ovarian Cancer Genes

**DOI:** 10.3390/cancers13133341

**Published:** 2021-07-03

**Authors:** Alejandro Moles-Fernández, Joanna Domènech-Vivó, Anna Tenés, Judith Balmaña, Orland Diez, Sara Gutiérrez-Enríquez

**Affiliations:** 1Hereditary Cancer Genetics Group, Vall d’Hebron Institute of Oncology (VHIO), Vall d’Hebron Barcelona Hospital Campus, 08035 Barcelona, Spain; amoles@vhio.net (A.M.-F.); jdomenech@vhio.net (J.D.-V.); jbalmana@vhio.net (J.B.); 2Area of Clinical and Molecular Genetics, Vall d’Hebron Hospital Universitari, Vall d’Hebron Barcelona Hospital Campus, 08035 Barcelona, Spain; atenes@vhebron.net; 3Medical Oncology Department, Vall d’Hebron Hospital Universitari, Vall d’Hebron Barcelona Hospital Campus, 08035 Barcelona, Spain

**Keywords:** spliceogenic deep intronic variants, pseudoexons, cryptic splice sites, splicing regulatory elements, hereditary breast ovarian cancer, in silico prediction tools

## Abstract

**Simple Summary:**

There is a significant percentage of hereditary breast and ovarian cancer (HBOC) cases that remain undiagnosed, because no pathogenic variant is detected through massively parallel sequencing of coding exons and exon-intron boundaries of high-moderate susceptibility risk genes. Deep intronic regions may contain variants affecting RNA splicing, leading ultimately to disease, and hence they may explain several cases where the genetic cause of HBOC is unknown. This study aims to characterize intronic regions to identify a landscape of “exonizable” zones and test the efficiency of two in silico tools to detect deep intronic variants affecting the mRNA splicing process.

**Abstract:**

The contribution of deep intronic splice-altering variants to hereditary breast and ovarian cancer (HBOC) is unknown. Current computational in silico tools to predict spliceogenic variants leading to pseudoexons have limited efficiency. We assessed the performance of the SpliceAI tool combined with ESRseq scores to identify spliceogenic deep intronic variants by affecting cryptic sites or splicing regulatory elements (SREs) using literature and experimental datasets. Our results with 233 published deep intronic variants showed that SpliceAI, with a 0.05 threshold, predicts spliceogenic deep intronic variants affecting cryptic splice sites, but is less effective in detecting those affecting SREs. Next, we characterized the SRE profiles using ESRseq, showing that pseudoexons are significantly enriched in SRE-enhancers compared to adjacent intronic regions. Although the combination of SpliceAI with ESRseq scores (considering ∆ESRseq and SRE landscape) showed higher sensitivity, the global performance did not improve because of the higher number of false positives. The combination of both tools was tested in a tumor RNA dataset with 207 intronic variants disrupting splicing, showing a sensitivity of 86%. Following the pipeline, five spliceogenic deep intronic variants were experimentally identified from 33 variants in HBOC genes. Overall, our results provide a framework to detect deep intronic variants disrupting splicing.

## 1. Introduction

Pathogenic variants in the tumor suppressor genes *BRCA1* and *BRCA2* (*BRCA1/2*) and other genes, mainly involved in DNA repair, have been linked to high or moderate risks of developing hereditary breast and ovarian cancer (HBOC) [1,2]. The identification of pathogenic variants in these genes offers patients and families precise clinical management based on personalized prevention and therapeutic strategies [3]. However, there is still a significant fraction of cases for which the genetic analysis does not identify causative variants underlying their predisposition to breast and/or ovarian cancer [4,5,6].

Currently, the detection of pathogenic variants is addressed mainly by massively parallel sequencing of high-moderate penetrance gene panels. An important number of identified deleterious variants affect pre-mRNA splicing and interestingly, hereditary cancer genes (including some HBOC and Lynch syndrome genes) are enriched for this type of variants [7]. The spliceogenic variants may occur in both introns and exons and disrupt consensus “*cis*” sequences such as canonical splice site nucleotides, branch point, polypyrimidine tract motifs, and Splicing Regulatory Elements (SREs). SREs are sequences that act as splicing intronic/exonic enhancers (ISE/ESE) or silencers (ISS/ESS), binding (SR)-rich proteins and heterogeneous nuclear ribonucleoproteins (hnRNPs), respectively [8,9]. Exon definition is the initial step in pre-mRNA splicing, and it has been suggested that accurate splice site recognition resides in a SRE balance, i.e., exons enriched with Exonic Splicing Enhancers (ESEs) and introns with Intronic Splicing Silencers (ISSs) [10,11,12].

In contrast to several studies showing the spliceogenic effect of exonic variants, there is a lack of information about the frequency of deleterious variants occurring in deep intronic regions (±20 bp from canonical splicing sites) since conventional genetic diagnosis is usually restricted to coding exons and flanking intronic regions. However, nucleotide changes in these regions could generate aberrant transcripts by introducing intronic sequences in mature mRNA [13,14,15]. In fact, pathogenic deep intronic variants have been described in more than 75 disease-associated genes including monogenic disorders such as hereditary cancer syndromes [16]. Deep intronic variants can alter splicing by two different mechanisms [17]: the creation/enhancement of cryptic splice sites and the alteration of an intronic SRE by the disruption of an ISS or the creation/strengthening of an ISE (Figure 1). A few examples of diseases driven by the inclusion of a pseudoexon due to these phenomena are HBOC caused by the c.4185 + 4105C > T variant in *BRCA1*, the first reported deep intronic variant in this gene that activates a pre-existing cryptic donor site [18], and the Ataxia-telangiectasia disease due to the c.2839-581_2839-578del variant in the *ATM* gene, which creates an ISE [19]. These examples highlight the relevance of screening deep intronic regions in HBOC patients to identify germline pathogenic variants leading to an aberrant RNA processing.

Given that experimental testing of all possible spliceogenic detected variants is currently not feasible in a clinical setting, multiple computational prediction tools have been developed to infer their effect and hence to prioritize variants to be experimentally evaluated [20]. Moreover, computational predictions of splicing variants are part of the supporting evidence included in the variant interpretation guidelines of the American College of Medical Genetics and Genomics (ACMG) [21]. Although different in silico tools have been published that accurately identify splicing exonic variants affecting canonical splice sites (MES, SSF, HSF) [21,22,23,24] or altering SREs (ESRseq, HZ_EI_, HAL) [11,25,26,27], there has been limited success in identifying deep intronic variants [18,27,28].

Deep learning tools such as SpliceAI, could outperform classical prediction approaches [29], but little is known about its performance for identifying deep intronic variants affecting splicing either by creating de novo/enhancing splice sites or SREs [27,28]. Our work assesses the performance of the SpliceAI tool combined with ESRseq scores to identify spliceogenic deep intronic variants using literature and experimental datasets.

Moreover, the arrangement of *cis* splicing elements in deep intronic regions, especially those corresponding to regulatory elements, would configure sequences with structures similar to those of canonical exons [30,31]. This landscape of regulatory zones would favor the generation of pseudoexons if a new deep intronic variant helps to define these structures. Thus, in this study we also compare the SRE presence between canonical exons of HBOC genes and published pseudoexons in order to characterize an “exon-like landscape” across introns which would help to identify potentially “exonizable” intronic regions.

To our knowledge, we provide the first in silico framework to prioritize deep intronic variants for their experimental RNA analysis, taking into account the landscape of splicing elements and highlighting the importance of SRE balance in the inclusion of sequences in mature RNA.

## 2. Materials and Methods

### 2.1. Datasets

Four datasets were used in this study: (i) 233 blood-detected germline deep intronic variants (located >20 nt from known exon-intron boundaries) collected from literature using the keywords “pseudoexon” and “deep intronic” (Appendix A). Their splicing effect had been assessed experimentally by gel electrophoresis or Sanger sequencing using blood, lymphoblastoid cell lines, midigenes, or minigenes. We defined as deep intronic variants those that were more than 20 nt away from the nearest exon-intron junction because these nucleotides are outside of the known splice site consensus sequences [8,9]; (ii) 1161 exonic variants compiled by Tubeuf et al. [27] to compare the accuracy of SREs-dedicated algorithms for predicting splicing alteration by affecting SREs (Appendix A); (iii) an additional supportive dataset of 207 somatic deep intronic splice-altering variants detected and characterized in tumor by both whole-genome sequencing and RNA-sequencing, retrieved from Jung et al. [32] (Appendix A); and (iv) an experimental dataset from patients of Hospital Universitari Vall d’Hebron, comprised of 33 unique intronic germline variants and selected according to their blood RNA availability (Appendix A). The splicing impact of all 33 variants was assessed in silico with SpliceAI and ESRseq and experimentally characterized using whole blood RNA.

Additionally, we retrieved the sequences of all exons and adjacent 100 intronic nucleotides of HBOC and Lynch genes (*BRCA1, BRCA2, PALB2, BRIP1, RAD51C, RAD51D, PTEN, TP53, CDH1, CHEK2, BARD1, STK11, MSH2, MSH6* and *MLH1*) according to their NCBI reference transcripts (Appendix A), to compare the splicing regulatory balance between canonical exons and pseudoexons from the literature dataset. Taking into account that most exons contain less than 300 nt, and that the mean length is 147 nt [33], exceptionally long exons (e.g., exons 10 of *BRCA1* and 10 and 11 of *BRCA2*) were not included.

### 2.2. In Silico Variant Annotation and Analysis for All Datasets

SpliceAI (v1.3; https://github.com/Illumina/SpliceAI accessed on 1 January 2021) was run to obtain the delta (∆) score of a variant (DS), defined as the maximum of DS for acceptor gain (DS_AG) and DS for donor gain (DS_DG) for deep intronic variants and the maximum of DS for acceptor loss (DS_AL) and DS for donor loss (DS_DL) for exonic variants [34]. The DS value ranges from 0 to 1 and can be interpreted as the probability of a variant being splice-altering. We considered the SpliceAI estimations of a gained/lost splice site in the 4999 nucleotides located on each side of the variant.

For the prediction of variant-induced SREs alterations, the ∆ESRseq value was calculated according to Ke., et al. [11] as the difference between the ESRseq scores of a variant sequence of 11 nucleotides (5 nucleotides at each side of the variant) and wild type sequence scores (ESRseq VAR- ESRseq WT).

To get the SRE distribution along any genomic region, we focused exclusively on the ESRseq scores, which were also calculated according to Ke., et al., obtaining individual nucleotide scores of each one of the positions of a sequence [11] (Appendix A). The sum of the scores for each nucleotide in a given sequence was defined as “area”. To account for the differences in size between exons, the area was divided by the number of nucleotides of each exon obtaining a normalized SRE area value (Normalized SRE area = ∑ ESRseq scores of all nt of region of interest/size of region of interest). This value was used to compare the SREs of all constitutive exonic sequences from HBOC and Lynch genes, with all pseudoexons collected in the literature dataset.

Alamut Visual software v.2.10 (Interactive Biosoftware) was used for annotation of variants included in the patients’ experimental validation dataset, providing data of allele frequencies in general population from the Genome Aggregation Database (gnomAD 2.1) and variant classification reported in ClinVar database (https://www.ncbi.nlm.nih.gov/clinvar/ accessed on 1 March 2021), considering the number of variant submitters and reviews by expert panel.

### 2.3. Statistical Analysis

Performance values of sensitivity, specificity, overall accuracy, Positive and Negative Predictive Values, False and Positive Discovery Rates, and Matthews correlation coefficient (MCC) were calculated with different in silico tool thresholds individually and in sequential combinations. The statistical measures used for evaluation of the performance are depicted in Appendix A.

The SpliceAI (4999 bp) optimized threshold was calculated based on the highest MCC using the literature dataset. In the case of ESRseq, the cut-off optimization was estimated by maximizing the sum of specificity and sensitivity since in the optimization based on MCC, the sensitivity/specificity was too unbalanced.

Analysis of variance (ANOVA) and T-test were used to compare the means of SRE scores between canonical exons, adjacent intronic sequences and pseudoexons with their adjacent sequences. *T*-test was used to compare the absolute SRE differences between the group of variants causing pseudoexons and variants without any effect. All tests were performed using Graphad Prism 6.

### 2.4. Experimental RNA Analysis in Patient’s Data Set

The patient dataset included unrelated cases from HBOC families ascertained through the Familiar Cancer Unit of Hospital Universitari Vall d’Hebron, HUVH (Barcelona, Spain). A total of 33 unique intronic germline variants were selected based on blood RNA availability in *ATM*, *BARD1*, *BRCA2*, *FAM175A*, *MLH1*, *MSH2*, *MUTYH*, *NF1*, *PTEN*, *RAD51C*, *RBBP8,* and *TP53* (Appendix A). All variants were identified in DNA by massively parallel sequencing using Illumina technology with a diagnostic routine panel of coding exons and exon-intron boundaries or by a research panel specifically designed to sequence whole intronic regions and confirmed by Sanger sequencing [4,5]. Healthy individuals without familial cancer history were included as negative controls.

#### 2.4.1. Reverse Transcription-PCR (RT-PCR) and Sanger Sequencing

Total RNA from variant carriers and controls was isolated from 10 mL of peripheral blood using Trizol reagent (Invitrogen, Waltham MS, USA) following the manufacturer’s protocol. RNA was cleaned-up using RNeasy Mini Kit (QIAGEN, Hilden, Germany) with an additional step of DNase digestion using RNase-Free DNase Set (QIAGEN) or Ambion™ DNase I RNase-free (ThermoFisher, Waltham, MS, USA) in samples with a low RNA concentration. A total of 100 ng of RNA were retrotranscribed to yield cDNA using PrimeScript RT reagent kit (Takara Bio, Shiga, Japan), combining random and oligo-dT primers. PCR primers were designed to amplify a whole exon upstream and downstream from the intron containing the variant of interest. PCR assays were performed in 25 uL reaction volume containing 50 ng of cDNA as template, using BioTaq DNA Polymerase (Meridian Bioscience, Cincinnati OH, USA). Samples were denatured at 95 °C for 10 min, followed by 35 cycles consisting of 95 °C for 30 s, 56–62 °C for 30 s, and 72 °C for 1–7 min; and a final extension step at 72 °C for 7 min. All primers used in this study and amplification conditions are detailed in Appendix A.

Capillary electrophoresis using the 4200 TapeStation device (Agilent, Santa Clara CA, USA) with High Sensitivity D1000 ScreenTape reagents (Agilent) was used to assess the quality of PCR products. These products were enzymatically cleaned using ExoSAP-IT^−^ PCR Product Cleanup (Affimetrix, ThermoFisher Scientific, Waltham, MS, USA) and bidirectionally sequenced using BigDye Terminator v3.1 Cycle Sequencing Kit (Applied Biosystems, Waltham, MS, USA). Sequencing products were run in an ABI3130xl Genetic Analyzer (Applied Biosystems) and were analyzed using Sequencing Analysis v6.0 software (Applied Biosystems). The reference transcripts based on GRCh37 (hg19) genome and listed in the Appendix A were used for sequence alignment and transcript annotation.

#### 2.4.2. Qualitative Analysis by Capillary Electrophoresis of Fluorescent Amplicons

RT-PCRs using primers labelled with 6-Carboxyfluorescein (6-FAM) at the 5′ end were performed in triplicate (see labelled primers in Appendix A). These fluorescent products were assessed by high-resolution capillary electrophoresis to detect and annotate all amplified transcripts. Specifically, 0.5 µL of the PCR products were run in an ABI3130xl Genetic Analyzer instrument (Applied Biosystems) for fragment analysis. GeneScan 500 and 1000 ROX (Applied Biosystems) was used as internal size-standard. Electrophoresis conditions were the same for all samples: 60 °C, 12 s injection at 1.2 KV and 2000 s run at 12 KV. Data visualization and peak size-calling was performed using GeneMapper software v5.0 (Applied Biosystems).

The maximum fragment size that could be detected was 946 bp, using the internal size-standard 1000 ROX.

### 2.5. Editorial Policies and Ethical Considerations

This study was approved by the Clinical Research Ethics Committee (CEIC) of Hospital Universitari Vall d’Hebron, Barcelona, Spain. All individuals received genetic counseling and signed written informed consent for HBOC panel genetic testing and research studies.

## 3. Results

### 3.1. SpliceAI Optimally Predicts Deep Intronic Splice-Altering Variants but with Less Sensitivity Those Affecting Splicing by Altering Regulatory Elements

To establish the performance of SpliceAI in predicting deep intronic pseudoexon-generating variants, we interrogated a set of variants collected from the literature, after searching for variants located beyond 20 nucleotides from exon-intron boundaries and for which RNA data was available (Appendix A). This collection contains 233 deep intronic variants from 80 HBOC and other rare mendelian disease genes, including 133 variants that promote the creation of pseudoexons or intron retention events and 100 variants that do not alter splicing. Once the delta (∆) score for each variant was obtained by running SpliceAI (v1.3), it was compared with the experimental results in RNA to estimate sensitivity, specificity, and accuracy. Then, we estimated the threshold at which the performance of the tool was more optimal, obtaining at ∆ score of 0.05 the highest MCC of 0.86 (Table 1 and Appendix A).

Since SpliceAI was specifically developed to detect altering splicing variants by activating or creating cryptic splice sites [34], we reassessed its predictions separately considering two groups of intronic-splice altering variants according to the *cis* element affected. We obtained better results with the group of cryptic splice variants than with the SRE disruptive variants (0.88 vs. 0.66 of MCC, respectively; Table 1 and Appendix A).

The lower sensitivity showed by SpliceAI in predicting the impact of only 16 deep intronic variants on SREs prompted us to assess its performance with a large previous published dataset, composed of 360 exonic variants that affect splicing by altering SREs and 801 exonic variants without effect on splicing (Appendix A) [27]. The performance with our pre-established 0.05 cut-off was 69.16% sensitivity, 84.27% specificity and 0.53 MCC, while with the cut-off of 0.06 showing the highest MCC the prediction improves, reaching 0.548 MCC (Appendix A).

To supplement the performance of SpliceAI in identifying deep intronic variants disrupting SREs, we consecutively added the ESRseq evaluation. ESRseq is a computational algorithm specifically developed to predict SREs disruption that showed the best performance in predicting both variant-induced exon skipping and exon inclusion in a recent benchmarking study [27]. To do this, the ∆ESRseq values (differences between wild type (WT) ESRseq score and variant ESRseq score) were calculated as described in Ke et al. [11] for intronic variants negatively predicted by SpliceAI (<0.05). The threshold optimization by maximizing the sum of specificity and sensitivity, indicated that variants with a score change equal or higher than 0.63 were predicted to promote pseudoexon inclusion by altering SREs with higher sensitivity and specificity (Appendix A). The performance of the sequential pipeline applying to those variants with a SpliceAI ∆ score of <0.05 and the ∆ESRseq threshold of ≥0.63, showed higher sensitivity (96.24%) than those obtained by SpliceAI alone, but lower specificity (69%) and MCC (0.69) (Table 2). We next sought to know if the optimized SpliceAI cut-off of 0.05 with the whole set of variants, regardless of the *cis*-element affected, would be higher if they were evaluated using ESRseq primarily, i.e., SpliceAI would exclusively compute for variants affecting cryptic sites. However, this analysis of assessing the 233 intronic variants firstly with ESRseq and then with SpliceAI, led to the same previous optimized SpliceAI threshold of 0.05. 

### 3.2. Splicing Regulatory Elements Balance Is Similar between Pseudoexons and Canonical Exons

To further investigate the role of splicing regulatory elements (SREs) in RNA included sequences, we compared the SREs landscape between constitutive exons and pseudoexons.

First, we extracted, from canonical transcripts of HBOC and Lynch genes (Appendix A), the respective constitutional exon sequences and the 100 adjacent intron nucleotides for each gene. Next, an ESRseq value was assigned to each nucleotide according to Ke et al. [11], thus obtaining a map of the distribution of regulatory elements along the different exons. In addition, we calculated the sum of the total values (area), and the normalized SRE area score (sum of total scores/number of region length nucleotides), of each exonic and adjacent intronic region for each gene. In Appendix A, we show an example of the obtained data for *BARD1* gene; data for all other genes is available upon request. Comparing the values obtained from canonical exons and adjacent upstream and downstream introns, significant differences were observed. Exons were enriched in positive values, corresponding to exonic splicing enhancers (ESE), while in intronic regions predominated the negative values, indicating an abundance of intronic splicing silencers (ISS) (Figure 2A).

We then determined the presence and proportion of SREs in the pseudoexons in our literature dataset. The sequences of the pseudoexonized regions or intron retentions and the adjacent up and downstream 100 nucleotides were analyzed, following the cDNA position indicated in the corresponding publication (Appendix A). These sequences were mapped with ESRseq scores obtained for each nucleotide, thus obtaining the area and the normalized SRE area values. Next, we compared the ESRseq values obtained between the region included in the mRNA as a pseudoexon and in upstream and downstream intronic regions. As with the canonical exons, significant differences were also observed. The regions included in the mRNA as a pseudoexon due to the variant presented a higher percentage of positive values and for those that remained as introns, a greater proportion of negative values (Figure 2B; data in Appendix A).

However, these differences were more pronounced between canonical exons and their surrounding intronic regions (Figure 2C), since the presence of enhancer or silencer regulatory elements is more abundant in exonic and intronic canonical regions respectively (Appendix A).

We also estimated for each variant of the literature dataset, regardless of its splicing effect, the ESRseq values for each position of the 100 intronic bases before and after the variant (without including the hexamers affected by the variant), and then the absolute difference of normalized SRE area between these two regions was calculated (Appendix A). This value was used to compare the two groups of deep intronic variants: 133 spliceogenic vs. 100 no effect variants. Our results demonstrated that variants with no effect presented a very low difference between adjacent regions, compared to variants with a spliceogenic effect (Figure 3). This is consistent with spliceogenic intronic variants being in regions with a significant difference in the balance of SRE between exons and introns (in line with the pseudoexon landscape showed in Figure 2B), while variants that do not cause alteration are in intronic regions with no SREs that could make them susceptible to be pseudoexonized. Overall, these results suggest that: (i) intronic regions with a similar SRE balance to that of exons are more susceptible to be included in mature RNA; (ii) a SRE balance is relevant in the RNA misplacing caused by deep intronic variants and; (iii) bearing in mind this balance will facilitate the in silico identification of intronic variants leading to pseudoexon inclusions.

### 3.3. Inclusion of SRE Landscape in the In Silico Detection of Deep Intronic Splice-Altering Variants

Although SRE in silico tools (such as ESRseq) can detect variant-induced SRE alterations, they are not able to identify whether the SRE landscape where the variant is located presents similarities to those of an exon, and ignore the relevance of the SRE balance to identify variant-induced pseudoexon events. To address this limitation, we included in the pipeline of SpliceAI (∆ score cut-off of </≥0.05) and ESRseq (∆ score cut-off of +0.63), the estimation of the absolute difference between 100 nucleotides up and downstream for each variant (Appendix A). With the absolute difference threshold of 0.51 we obtained a MCC of 0.83, with 95.49% sensitivity, 86% specificity, and 91.42% accuracy (Table 2 and Appendix A). Although this pipeline would theoretically detect variants that alter SREs, the sensitivity of 95.49% only slightly improves that observed using SpliceAI alone (93.98%) (Table 2).

The performance of the last pipeline (SpliceAI ∆ score 0.05 -> ∆ESRseq 0.63 -> difference in absolute values 0.51), was tested with a set of 207 splicing-disrupting deep intronic variants identified in tumors by RNAseq [32] with 86% sensitivity which again slightly improves that observed using SpliceAI alone (85.5%) (Appendix A).

### 3.4. Experimental Analysis of Hereditary Cancer Gene Variants

Thirty-three unique variants were experimentally assessed. Thirteen variants passed the 0.05 SpliceAI threshold and six presented ∆ESRseq equal or greater than 0.63 and an absolute difference value greater than 0.51 (Appendix A). The remaining 14 variants were not predicted as spliceogenic (Appendix A). We characterized the variant effect by RT-PCR assays comparing their splicing profiles (by high-resolution electrophoresis) with those in the healthy controls, and posterior Sanger sequencing. This analysis detected the inclusion of intronic regions in mature mRNA in only 5 of the 13 variants prioritized by SpliceAI alone (Table 3 and Figure 4): *ATM* variants c.1899-123A > G, c.2466 + 1552G > C, c.8850 + 2029A > G, *FAM175A* variant c.476 + 158G > T and *MUTYH* variant c.998-27G > A. The six variants with a ∆ESRseq equal or greater than 0.63 and absolute difference prediction in favor of altering a SRE region did not show an aberrant splicing. All 14 variants with a negative splicing alteration prediction presented a normal splicing pattern. Table 4 shows that sensitivity and specificity using SpliceAI alone for all variants with the cut-off of ≥0.05 is higher than the pipeline of applying ∆ESRseq equal or greater than 0.63, and an absolute difference value greater than 0.51 for those variants with ∆ SpliceAI scores of <0.05.

## 4. Discussion

The contribution of deep intronic variants to HBOC disease is not well known due to their location in poorly screened regions, but their potential effect on transcript splicing including intron sequences in mature RNA may be clinically significant [10,26]. For this reason, the identification and subsequent RNA characterization of this type of variants should be considered when conventional genetic analysis focused on coding regions and exon/intron boundaries does not lead to the identification of pathogenic variants [35].

However, the identification of deep intronic variants is challenging due to the lack of specific in silico pipelines [28]. Recently, some published studies using SpliceAI, a deep learning-derived algorithm [34], suggest its utility to identify with high efficiency intronic and exonic variants creating or enhancing cryptic splice sites and leading to splicing alterations [36,37,38,39]. Nevertheless, to our knowledge, only small datasets of deep intronic variants have been used to test the performance of the SpliceAI tool for identifying this type of variants [27,28,34,37].

Our work provides a large dataset of deep intronic variants that are clinically relevant, as they were tested in a clinical setting using blood, mini, or midigene assays and is well balanced with 133 altering and 100 non-altering splicing events. Hence, this data can be used as a positive control training set for further improvements of computational prediction tools. With this data, we confirmed that SpliceAI with a threshold of ≥0.05 has an optimal predictive value in the identification of spliceogenic deep intronic variants, obtaining a MCC of 0.86. Interestingly, Riepe et al. [37] with an optimized SpliceAI cut-off score of 0.18, also showed a high performance of 0.84 MCC for predicting 81 deep intronic variants in the *ABCA4* gene, that are also included in our literature dataset. The authors further demonstrated that SpliceAI was the best tool for these 81 deep intronic variants compared with other deep-learning based algorithms [37]. In this line, Jaganathan et al. [34] in their SpliceAI development article, demonstrated that by applying SpliceAI with a cut-off of ≥0.5 to GTEx RNAseq data, it achieved a sensitivity of 71% when the variants were near exons (82 variants, overlapping exons or ≤50 nt from exon-intron boundaries), but dropped to 41% when the variants were in deep intronic regions (37 variants, >50 nt from exons). In sum, our study together with the two last works mentioned above, support that a low SpliceAI threshold is needed to especially detect deep intronic splice-altering variants. In our experimental dataset with clinical variants, using our optimized ≥0.05 threshold, SpliceAI attained a performance of 0.62 MCC, predicting all five experimentally confirmed spliceogenic variants, but with a low specificity of 71.42% due to a high number of false-positives (Table 4).

Besides creating or enhancing cryptic splice sites, the intronic variants can lead to the inclusion of pseudoexons by creating or disrupting intronic SREs. Our evaluation of the prediction of splice alteration through SRE involvement of both deep intronic (Table 1 and Figure 1) and exonic (Appendix A) variants pointed out that for SpliceAI, it is more challenging to predict the impact of this type of variants. This is possibly due to the fact that the deep learning network approach used for SpliceAI development was not able to account for the SREs, denoting that the performance of SpliceAI can still be improved. To note, this is the first study testing the prediction capacity of a deep learning method such as SpliceAI of exonic variants disrupting SREs leading to exon skipping.

To supplement lower SpliceAI performance for detecting SREs altering variants, we added the ESRseq, which has a high capacity to recognize this type of variants [27], obtaining an increase of sensitivity but a lower specificity (Table 2). We reasoned that this limitation was due to the fact that ESRseq evaluates on a hexamer local level, without accounting for a SREs landscape that defines a region to be included as a pseudoexon. This prompted us to characterize the landscape of SREs in pseudoexons using SRE scores, showing that the relation of the SRE landscape between the pseudoexon and flanking introns is similar to that of canonical exons, but less defined (Figure 2C and Appendix A). In contrast, the ESRseq developers in Ke et al. [11] reported that the pseudoexons did not present a different balance of SRE concerning the adjacent intronic regions. This discrepancy could be due to the fact that the pseudoexons analyzed in the above-mentioned work were theoretically defined, without an experimental RNA evaluation, as the intronic sequences had lengths between 50 and 250 nt and consensus values based on the Shapiro-Senapathy algorithm, of ≥75 for 3′ splice sites and ≥78 for 5′ splice sites, and were located beyond 100 bp from the exons [11]. Instead, we collected 133 pseudoexons from literature, experimentally confirmed using patient, mini, or midigene-derived RNA. Notably, similar findings to our results using approaches other than ESRseq tools have been recently reported, in 42 pseudoexons experimentally validated in the *DMD* gene, showing a smaller density of exonic splicing enhancers (ESEs) together with a higher density of exonic splicing silencers (ESSs) compared to canonical exons, which suggested that the pseudoexons presented a weaker exon profile in terms of SREs [30]. Interestingly, these differences have also been observed between alternative and canonical exons. The effect of variants altering the balance of SREs appears to be greater in alternative exons, which have fewer redundant enhancer elements, compared to constitutive ones [40]. Therefore, we suggest that deep intronic variants that strengthen an enhancer or even decrease a silencer will have a greater chance of being spliceogenic provided they are located in intron regions with an exon-like SRE landscape, similar to what happens in alternative exons.

Given the role of an exon-like SRE landscape in the inclusion of pseudoexons, we combined the ∆SpliceAI 0.05 and 0.63 ∆ESRseq optimized thresholds, and the absolute difference of SRE balance between the regions before and after a variant. This last value, with the variants compiled from literature, indicates clear differences between splicing altering variants and those without any effect (Figure 3) and helps to identify spliceogenic variants, with high sensitivity and specificity (Table 2). The combination was assayed in a tumor RNA-seq dataset (Appendix A) [32] obtaining a sensitivity of 86%. Moreover, applying the same in silico combination to a set of deep intronic variants from a cohort of HBOC patients allowed us to identify five spliceogenic variants that were predicted by SpliceAI to activate a cryptic splice site nearby, while assaying eight false positives (Appendix A).

The low accuracy of the in-silico strategy combining ∆SpliceAI, ∆ESRseq, and SRE landscape using ESRseq scores can be explained by two reasons. First, our literature dataset only contains 16 splice altering variants by affecting a regulatory element. Second, the splicing aberrations of the literature and experimental datasets were assayed using RT-PCR, which has an intrinsic bias towards smaller amplicons. This is especially relevant in the case of pseudoexons as they generate larger fragments than normal transcripts and are also less expressed if they cause a premature termination codon and are targeted by non-sense mediated decay. Massively long-read RNA sequencing, such as those using Oxford Nanopore Technologies, could address this limitation by allowing simultaneous detection and quantification of all RNA transcripts avoiding the PCR amplification step [41].

Additionally, it is worth stating that the main purpose of this work was not the clinical classification of the variants assessed but rather to investigate how to improve the in silico identification of deep intronic splicing variants. Qualitative analysis can only detect aberrant transcripts, but additional quantification of the functional transcripts, co-segregation data or other functional assays are needed to classify the variants that induce pseudoexons [42,43].

Overall, the results obtained with the three sets of deep intronic variants (literature, tumor, and experimental) demonstrated that SpliceAI alone is able to identify variants causing pseudoexons and that the addition of ESRseq increases the number of false positives. Moreover, our use of ESRseq values to map the SRE balance in canonical exons and pseudoexons differentiating exonic landscapes from intronic ones suggest that this approach might be systematically used to identify exon-like landscapes in introns of HBOC and Lynch genes, thus helping to interpret whether an intronic variant makes a region much more exonizable.

## 5. Conclusions

We have provided evidence that SpliceAI, a deep learning-based in silico tool, can predict splicing altering deep intronic variants with high-performance. However, its accuracy is limited with variants affecting SREs, either with intronic variants introducing pseudoexons or exonic variants inducing exon skipping. The addition of ESRseq, a specific bioinformatic tool to detect SRE disruption/enhancement, did not increase the accuracy of the deep intronic splicing-altering variants prediction. However, our findings show that pseudoexons have a “SRE landscape” similar to that of exons. This indicates that intronic regions with a high potential to be included as pseudoexons can be systematically identified throughout the HBOC genes, facilitating the in silico detection of spliceogenic deep intronic variants.

## Figures and Tables

**Figure 1 cancers-13-03341-f001:**
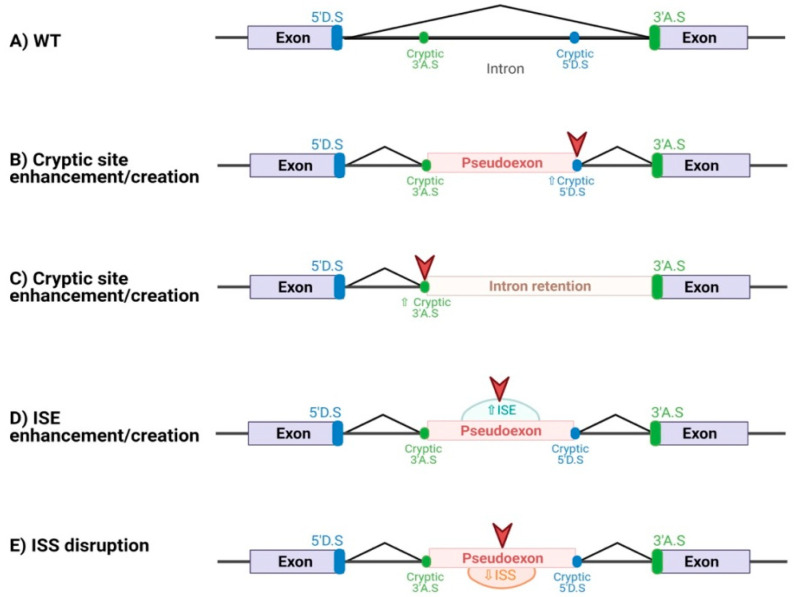
Splicing effects caused by deep intronic variants. (**A**) Normal splicing using natural splicing sites. (**B**) Deep intronic variant creating/enhancing a cryptic splice site, resulting in the inclusion of a pseudoexon by using a complementary cryptic site. (**C**) Intronic retention caused by a deep intronic variant that creates/enhances a cryptic site, which is used instead of the canonical splice site. (**D**) Deep intronic variant creating/enhancing an ISE, resulting in the inclusion of a cryptic exon using two cryptic splice sites. (**E**) Deep intronic variant disrupting an ISS, resulting in the inclusion of a cryptic exon using two cryptic splice sites.

**Figure 2 cancers-13-03341-f002:**
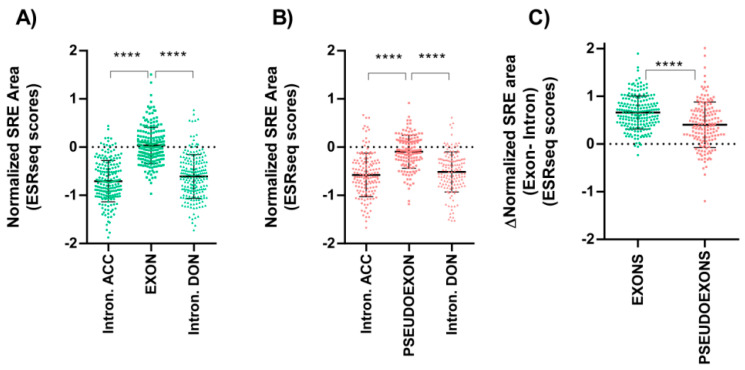
Definition of SREs abundance in different genomic regions, using the normalized SRE area calculated with ESRseq scores. (**A**) Normalized SRE area using ESRseq scores of exonic and adjacent 100 intronic nucleotides located upstream and downstream of canonical exons of HBOC and Lynch genes. Significant differences were identified between exons and intronic regions (Pair-wise significance levels calculated by Tukey test, **** *p*-value < 0.0001). (**B**) Normalized SRE Area using ESRseq scores of pseudoexons and adjacent 100 intronic nucleotides located upstream and downstream of pseudoexons listed in the literature dataset. Significant differences were identified between pseudoexons and intronic regions (Pair-wise significance levels calculated by Tukey test, **** *p*-value < 0.0001). (**C**) Comparison of the exon-intron difference of normalized SRE areas between canonical exons and pseudoexons. First, the mean of normalized SRE area of adjacent donor and acceptor site intronic regions was calculated. Then, this mean was subtracted from the exon and pseudoexon normalized SRE area value. The difference between exonic and intronic regions in canonical exons was higher than in the case of pseudoexons, suggesting that they are more defined by a SRE balance. (*t*-test, **** *p*-value < 0.0001). Mean ± standard deviation is represented in each graph. Intron. ACC: intronic sequence adjacent to acceptor site; Intron. DON: intronic sequence adjacent to donor site.

**Figure 3 cancers-13-03341-f003:**
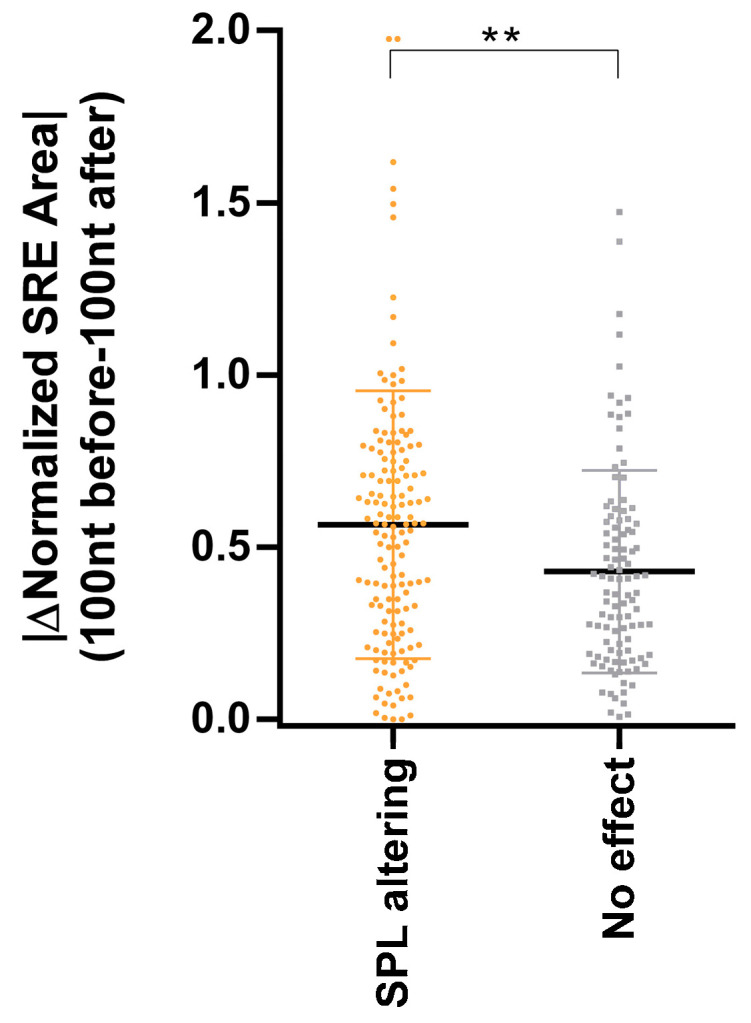
Comparison of the absolute difference of normalized SRE area from 100 nucleotides before and after each deep intronic variant compiled from the literature. Absolute values of normalized SRE area difference from 100 nucleotides upstream and downstream of each variant were used to compare those spliceogenic with those without any effect. Splicing variants (SPL altering) showed higher differences between previous and posterior sequences (*t*-test, ** *p*-value ≤ 0.01). Mean ± standard deviation is represented.

**Figure 4 cancers-13-03341-f004:**
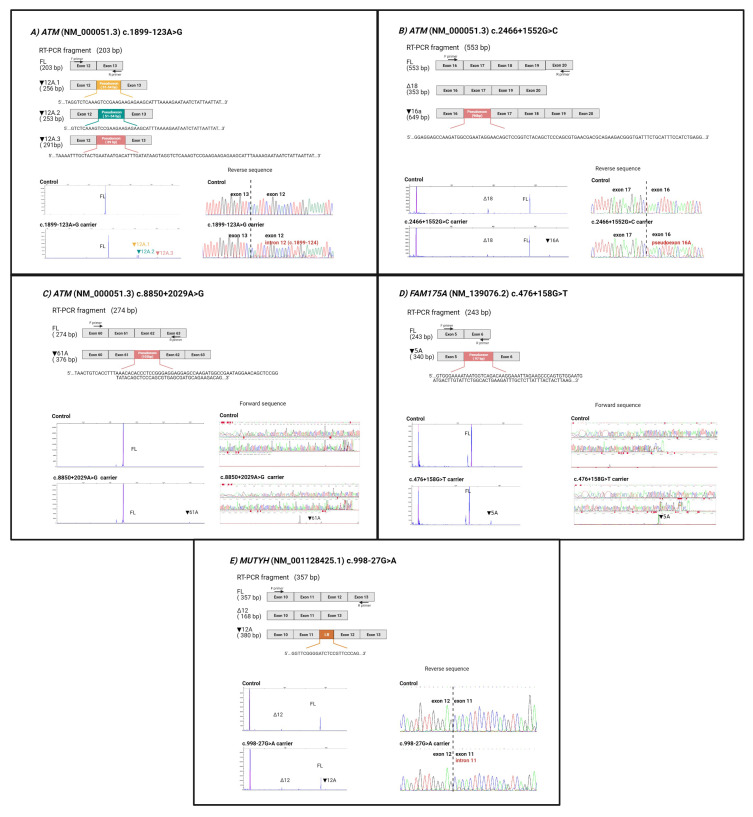
Spliceogenic variants characterization in patients’ RNA. For each variant, there is an RT-PCR assay graphical representation, the results of capillary electrophoresis of 6-FAM labelled amplicons and Sanger sequencing to confirm the expression of additional transcripts. (**A**) The *ATM* c.1899-123A > G variant activates a cryptic donor site, which is used to yield three different pseudoexons: ▼12A.1, ▼12A.2, and ▼12A.3, each generated as result of the usage of different cryptic acceptor sites (c.1899-174, c.1899-177, and c.1899-213) and the cryptic donor site created by the variant. The ▼12A.1 and ▼12A.2 transcripts were equally expressed, and their abundance was greater than the ▼12A.3. (**B**) The *ATM* c.2466 + 1552G > C variant generates the ▼16A additional transcript. This pseudoexon comprises nucleotides from the acceptor site created by the variant and the cryptic donor at c.2466 + 1650. (**C**) The *ATM* c.8850 + 2029A > G variant presents an additional transcript (▼61A), from the cryptic acceptor site created by the variant to the cryptic donor at c.8850 + 2131. It was not possible to clearly read the sequence of the aberrant transcript because of its low expression levels, but in the Sanger sequence, we could detect the additional transcript with the insertion since it is marked by the FAM signal at the end of the fragment. (**D**) The *FAM175A* c.476 + 156G > T variant leads to the inclusion of a pseudoexon (▼5A), which results in the usage of the cryptic acceptor site activated by the variant and the cryptic donor at c.476 + 252. Its abundance was very low, but the transcript with the insertion was also detected in the Sanger sequence since it is marked by the FAM signal at the end of the fragment. (**E**) The *MUTYH* c.998-27G > A variant creates/enhances a cryptic acceptor site which is used instead of the natural acceptor site of exon 12, generating an intronic retention (▼12A transcript).

**Table 1 cancers-13-03341-t001:** Performance of SpliceAI for literature database with 233 deep intronic variants using the optimized threshold of 0.05.

Dataset	Splicing Altering Variants	No Splicing Altering Variants	Sensitivity	Specificity	Accuracy	MCC
All variants	133	100	93.99	92.00	93.13	0.86
Cryptic splice	117	100	95.73	92.00	94.01	0.88
SREs altering	16	100	81.25	92.00	90.52	0.66

**Table 2 cancers-13-03341-t002:** Prediction performances of SpliceAI alone and sequentially combined with ESRseq tool with 233 intronic variants from the literature database (133 altering and 100 non altering splicing). Abs. Dif.: absolute difference.

Pipeline	Sensitivity	Specificity	Accuracy	MCC
(1) ∆ SpliceAI ≥ 0.05	93.98	92.00	93.13	0.86
(2) ∆ SpliceAI ≥ 0.05 + SpliceAI < 0.05 and ∆ESRseq ≥ 0.63	96.24	69.00	84.55	0.69
(3) ∆ SpliceAI ≥ 0.05 + SpliceAI < 0.05 and ∆ESRseq ≥ 0.63 and Abs. Dif. 0.51	95.49	86.00	91.42	0.83

**Table 3 cancers-13-03341-t003:** Deep intronic variants with a spliceogenic effect detected in hereditary cancer genes.

Gene *	cNomenclature **	Intron	SpliceAI ^†^ (Position of Predicted Splice site)	∆ESRseq ^‡^	ABS dif.	Splicing Outcome ^§^	Population Variant Frequencies (GnomAD)	ClinVar Review Status ¥
*ATM*	c.1899-123A > G	12	**AG 0.15** (−51 bp) and **AG 0.74** (−90 bp)/**DG 0.71** (−1 bp)	**0.633**	**1.217**	Three pseudoexons: ▼12A.1 (r.1899_1900ins1899-174_1899-124), ▼12A.2 (r.1899_1900ins1899-177_1899-124) ▼12A.3 (r.1899_1900ins1899-213_1899-124)	0.000032	NR
	c.2466+1552G > C	16	**AG 0.93** (3 bp)/**DG 0.69** (97 bp)	−0.144	**1.816**	Pseudoexon ▼16A (r.2466_2467ins2466 + 1555_2466 + 1650)	NR	Likely pathogenic |1|
	c.8850+2029A > G	61	**AG 0.22**(1 bp)/**DG 0.16** (102 bp)	−0.187	**0.765**	Pseudoexon▼61A (r.8850_8851ins8850 + 2030_8850 + 2131)	NR	NR
*FAM175A*	c.476+158G > T	5	**AG 0.17** (2 bp)/**DG 0.22** (−94 bp)	0.559	**0.591**	Pseudoexon ▼5A (r.476_477ins476 + 156_476 + 252	0.000446	NR
*MUTYH*	c.998-27G > A	11	**AG 0.41** (−4 bp)/**DG 0.05** (−215 bp)	0.344	**1.000**	Intron retention ▼12A (r.997_998ins998-23 + 998)	0.001192	Likely benign|Likely benign 1|1

* Reference sequences for annotating variants: NM_000051.3 for *ATM*; NM_139076.2 for *FAM175A*; NM_001128425.1 for *MUTYH*. **HGVS nomenclature guidelines were used for variant annotation (http://varnomen.hgvs.org/ accessed on 1 September 2020). ^†^ SpliceAI prediction: AG = Acceptor Gain/DG = Donor Gain. The ∆Score range from 0 to 1 and can be interpreted as the probability that the variant affects splicing at any position around it. For each variant, SpliceAI evaluates a nucleotide window (+/− 4999 in this case) to see how the variant affects the probabilities of different positions in the pre-mRNA being splice acceptors or donors. The values in brackets represent positions with the biggest probability of being used as splicing sites within the window. Negative values are upstream (5’) of the variant, and positive are downstream (3’) of the variant. In bold are the scores of ≥0.05. ^‡^ ∆ESRseq score was calculated as described by Ke, et al., and in bold are indicated values ≥0.63. Absolute difference (ABS dif.): absolute value of the difference between the normalized SRE area of 100 bp before and after a variant. In bold are indicated values ≥0.51. ^§^ RNA splicing profiles were compared between carriers and controls by capillary electrophoresis and Sanger sequencing. ¥ Variant classification reported in ClinVar database (https://www.ncbi.nlm.nih.gov/clinvar/ accessed on 1 March 2021) according to number of variant submitters. NR: not reported.

**Table 4 cancers-13-03341-t004:** Performance pipeline (experimental variants dataset) with 33 variants: 5 splicing altering and 28 non-altering.

Pipeline	Sensitivity	Specificity	Accuracy	MCC
Splice ≥ 0.05	100	71.42	75.75	0.62
Splice ≥ 0.05; ESRseq ≥ 0.63; Abs. Dif. 0.51	100	50.00	57.57	0.43

## Data Availability

All data analyzed in this study are included in this article and its Appendix A.

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
