# Peer review of "Role of Splicing Regulatory Elements and In Silico Tools Usage in the Identification of Deep Intronic Splicing Variants in Hereditary Breast/Ovarian Cancer Genes"

_cancers, 2021, doi:10.3390/cancers13133341_

Round 1
Reviewer 1 Report
It is a well written study. And very important for translational medicine. Because it is really important to predict the effect of deep intronic variants.In most patients we can not find any pathogenic/likely pathogenic variants. They could have deep intronic variants and these variants can cause the cancer itself also.
Reviewer 2 Report
The manuscript suggests a pipeline for predicting the effect of deep intronic variants on RNA splicing.
First, the authors evaluated the SpliceAI tool alone. It satisfactorily detected cryptic splice sites, but not the variants affecting regulation of splicing in the training dataset (233 previously published variants). To improve the detection of variants affecting splicing regulation the authors suggested adding ESRseq-scores to SpliceAI. While the addition of ESRseq-scores marginally improved sensitivity, it significantly reduced specificity, so the overall performance was not better than SpliceAI alone (in both: the training set and in the authors’ own data of 33 deep intronic variants in HBOC genes). In an additional analysis the authors show that ESRseq-scores are higher in pseudo-exones than in other intronic regions, suggestive that although the value of adding ESRseq to SpliceAI was not confirmed by this study, it is logically sound and may be further investigated. Finally, the authors report 5 deep intronic variants causing splicing disruption in ATM, FAM175A and MUTYH genes.
The presented results could be interesting for readers involved in the interpretation of human sequencing data. However, several points need to be addressed before the publication:
Major points
1) The lack of improvement after addition of ESRseq to SpliceAI is not evident from the Abstract.
2) The authors describe the 207 deep intronic variants (Jung 2021) as a “validation” dataset. In fact, it can only be considered as an additional supportive resource because it doesn’t contain variants without effect on splicing. It is not possible to estimate sensitivity and specificity of classification on such a dataset. Therefore, the only true validation set is the authors’ own HBOC data, which is of very small size (just 33 variants). This should be made clear.
Minor points
3) In section 3.2 the authors found that pseudo-exons were enriched by splicing-enhancing-elements comparatively to the remaining areas of introns. At the same time, Ke et al 2011 reported that the pseudo-exons were not enriched by the splicing-enhancing-elements (e.g. see Fig 5 F in Ke et al 2011). The authors may wish to discuss this discrepancy, if they consider it relevant.
4) The 0.05 cutoff for SpliceAI is much lower than the conventional cutoff-s recommended by the SpliceAI authors (0.2 or 0.5 or even 0.8).This might deserve a discussion. Thus, Supp. Figure 1 provides support for such a low threshold when SpliceAI is used alone on the given data. However, it might be that a different threshold is optimal for the SpliceAI-ESRseq combination than for SpliceAI alone. Specifically, if the regulatory variants are detected primarily by ESRseq, then the threshold for SpliceAI could be selected using the cryptic sites only (which would bring it to a conventional value around ~0.2). This could be explored at the authors discretion.
Technical points
5) In section 2.3 the authors say that they used AUC-based optimization to select the threshold for delta-ERSseq. In fact, Supp. Table 9 suggests that the optimal threshold was selected by mere maximizing of the sum of sensitivity and specificity. This is also a valid (albeit simplistic) approach. However, it does not involve any AUC calculation. If the authors insist that they used some AUC, then they should explain it in more detail.
6) 1st paragraph in section 2.2 says that the authors used pre-computed SpliceAI data available on GitHub (does it include 4999 window data??). At the same time, line 236 on pg 6 says that scores were obtained by running Splice AI. Were the scores taken from pre-computed tables or by running the tool?
7) Notes to Figures 2A and 2B mention ANOVA for significance testing, while the figures themselves indicate pair-wise significance levels. This should be reconciled (I suppose that the pairwise significance levels were calculated by Tukey test or a similar method).
8) The authors should double-check formatting of supplementary tables, e.g. see irregularities on line 298 (columns AU-BA) of Supp. Table 2A
9) A fresh-eye proof-reading by a qualified native English speaker is needed to weed-out the language slip-ups, e.g. : “massively sequencing” should be “massively parallel sequencing” (2nd line of the abstract), “Sensibility” should be “Sensitivity” (table 2), etc. Also “enhancers” could be changed to “SRE-enhancers” in line 35 (pg1, Abstract) for clarity.
10) “Table 2” should be “Table 3” in the last line of section 3.1 (line 266 pg 6) and in section 3.3 (line 353 pg 9).
11) Reference 11 duplicates reference 26
Reviewer 3 Report
In this manuscript, the authors aimed to compare the performance of tools of in silico prediction of intronic variants that affect mRNA splicing. Overall, the manuscript is quite well written. If the authors could provide more rationales in justifying the study, it would be more interesting to the readers in Cancers. The current version may be more suitable to computational biology field. This could be improved if the issues as follows were addressed:
- What clinical implication is there in looking for splicing-related intronic variants? Some studies have shown splicing regulatory proteins associated with splicing signature and outcomes in cancer, e.g., CRC (PMID: 27650542), and intronic variants associated with gene expression levels, e.g., in ovarian cancer (PMID: 22822098).
- please define "deep intronic variants" as early as possible. is there any reference based on 20 nt as cutoff?
- How did the authors obtain SRE score, what method did the authors use?
- It seems that the authors only focus on germline intronic variants that affect splicing, and based on RNA-seq data could have somatic variants? How did the authors make sure they are not somatic mutations?
- It is also expected to clarify how to normalize SRE area in materials and methods.
- table 3, some more notations are helpful in interpreting the results, e.g., the column of "In", and the length of nucleotides in the column of SpliceAI.
Round 2
Reviewer 2 Report
The revised version of the manuscript has been improved; the authors have adequately addressed all my questions/suggestions.